# Application of Open Panicle Traits in Improving the Filling Characteristics at the Base of *Indica* Rice Panicles

**DOI:** 10.3390/plants13152035

**Published:** 2024-07-24

**Authors:** Guotao Yang, Qin Wang, Guoxing Yang, Guohao Zhang, Hong Chen, Xuechun Wang, Peng Ma, Yungao Hu

**Affiliations:** School of Life Science and Engineering, Southwest University of Science and Technology, Mianyang 621010, China; yangguotao2377893@163.com (G.Y.); wqying2022@163.com (Q.W.); 13953170122@163.com (G.Y.); 13541491443@163.com (G.Z.); chenhong_88@aliyun.com (H.C.); xuechunwang@swust.edu.cn (X.W.)

**Keywords:** *indica* rice, open panicle, rachilla angle, grain filling

## Abstract

Panicle-type structure is an important factor affecting rice yield, and an excellent panicle type has become a key indicator in rice breeding. In this study, the *indica* rice variety R766, which has an open panicle type, was obtained through natural mutation and hybrid selection. Through analyzing differences in panicle structure, panicle yield, and chemical composition between open panicle rice R766 and conventional panicle rice R2928, we found that the angles of the middle and lower branches in R766 were 186.99% and 135.93% greater than those in R2928, respectively. By comparing the grain-filling characteristics of different panicle positions in the two rice varieties, we found that the grain filling at the middle and lower panicle positions of R2928 was significantly lower, accompanied by an increase in the percentage of empty spikelets. However, in R766, the grain-filling rates in the middle and lower panicle positions were consistent with those in the upper panicle position, with significantly higher rates of grain filling and grain plumpness in the middle and lower panicle positions than in R2928. The empty grain rate at the lower panicle position of R766 was 15.25% lower than that of R2928, and the grain filling was 24.75% higher than that of R2928. Additionally, the variation in the 1000-grain weight of grains at different panicle positions in R766 was relatively small, with decreases of 1.55% and 0.38% in the middle and lower panicle positions, respectively, compared with the upper position, whereas R2928 showed decreases of 5.99% and 7.12% in the 1000-grain weight of grains at the middle and lower panicle positions, respectively, compared with the upper position. The cellulose content in the stems of R766 was 7.51% higher than that of R2928, with no significant difference in the cellulose content in the panicle axis and primary branches compared to R2928. The lignin content of the panicle axis in R766 was 8.03% higher than that in R2928, whereas there was no significant difference between the lignin content of the stems and primary branches. This preliminary study revealed the open panicle characteristics of R766 and the reasons for its high basal grain setting rate. This study provides a reference for promoting this open panicle-type *indica* rice variety to improve yield and disease resistance in environments with high humidity and low sunlight levels.

## 1. Introduction

In 1995, scientists of the International Rice Research Institute led by G S. Khush proposed the cultivation of “super rice”, which was later renamed the “new plant type” rice breeding program [1]. The plant type of crops depends on several aspects such as plant height, leaf type, number of tillers, tiller angle, and panicle morphology. Rice panicle type is an important component of rice plant-type breeding theory, and panicle traits are among the important indicators of panicle type. Panicle traits directly affect rice yield and quality [2,3]. So far, various classification methods for panicle types have been developed based on different panicle traits and research purposes. The panicle type refers to their mode of branching, the angle of the primary branches and the spikelet density. It can be simplified into three categories, compact, intermediate and open [4,5]. According to grain density, it is usually divided into scattered panicles, semi-scattered panicles, semi-compact panicles, and dense or compact panicles [6]. According to the number of grains on secondary branches and the distribution of primary branches on the rachis, rice varieties are divided into five types, i.e., lower dominant, upper dominant [6], middle dominant panicle type, slightly lower dominant, and slightly upper dominant panicle types [7]. Panicle types include erect-panicle types (less than 30°), semi-erect-panicle types (30–45°), and curved-panicle types (greater than 45°) based on the curvature of the panicle neck [8].

One researcher argued that the erect-panicle type can reduce the shading of the canopy, facilitate photosynthesis of the panicles, and provide population advantages [9]. The main reason for this is that during the filling and fruiting stages, the rice panicles remain upright, avoiding the interaction of shading between the rice panicles and the top three leaves, thereby improving the light-receiving conditions of the panicles and the top three leaves. During the rice-growing season in the hilly areas of Southwest China, there is limited sunlight, abundant rainfall, and high humidity, which can lead to severe rice diseases. Research has shown that the higher the humidity in the rice panicle, the more severe the occurrence of rice smut. Dense panicles, erect panicles, and other varieties with compact grains exhibit high humidity in the panicle layer and a high incidence of false smut. The occurrence of rice smut follows the pattern of erect-panicle type > semi-erect-panicle type > curved-panicle type [10]; however, curved-panicle varieties have low branch hardness and sparse grains. After heading, the panicles are bent. Although sparse grain density can reduce humidity, the bending of the panicle will lead to poor light transmittance of rice and less light radiation from the upper and middle leaves, thus affecting photosynthesis during the heading and filling period (which determines the key period for yield formation) [11].

To address the shortcomings of curved and erect panicle varieties, coordinate the distribution of population light resources, and reduce the occurrence of rice smut, the authors of this study propose a new approach of using open panicle rice (which has a large branch angle due to the growth of pulvinus at the base of branches [12]) to balance the high light transmittance of erect panicle types and reduce the humidity of the panicle layer. Therefore, the authors obtained open panicle-type germplasm resources through natural mutations and created open panicle-type *indica* rice R766 through hybrid backcrossing (Figure 1). At the filling stage, the air mobility of the R766 population was significantly higher than that of the curved-panicle type, and the incidence rate of false smut in rice was significantly lower than that in the erect-panicle and curved-panicle types. Simultaneously, the seed setting rate at the base of the curved panicle-type rice was significantly lower than that in the middle and upper parts.

This study intends to analyze the differences in panicle structure, panicle yield, chemical composition, etc., between open panicle rice R766 and conventional panicle rice R2928 to preliminarily elucidate the open panicle characteristics of R766 and the reasons for its high basal grain, and to provide a reference for improving the yield and disease resistance of rice in high-humidity and low-sunlight ecological environments through the promotion of open panicle-type *indica* rice varieties.

## 2. Results

### 2.1. Analysis of Phenotypic Traits of Rice Panicles

The panicle phenotype between open panicle rice R766 and normal panicle rice R2928 differed significantly. The angle between the middle and lower branches and rachis of R766 was significantly greater than that of R2928 (Figure 1A), and the panicle drooping curvature was significantly lower than that of R2928 in the upright state in the field (Figure 1B). Analysis of the specific phenotypes of the two panicle types (Figure 2) showed that the branch angle in the lower part of the R766 panicle was greater than that in the upper part, whereas there was little difference in the branch angle between the middle, lower, and upper parts of R2928. The angle of the branches in the lower part of the open panicle-type rice was significantly greater than that in the upper part, and the arrangement of the branches in the panicle was that of the open. The angles of the middle and lower branches of R766 were 186.99% and 135.93% higher than those of R2928, respectively. The diameters of the rachis and primary branches of R766 at different positions were significantly larger than those of R2928. The rachis diameters were 17.86% (upper part), 20.73% (middle part), 17.24% (lower part), and 17.48% (1 cm below the rachis) larger than those of R2928. The diameters of the primary branches at the different positions were 33.33% (upper part), 50.00% (middle part), and 58.97% (lower part) larger than those of R2928. The diameters of the middle and lower branches of the different panicle types of rice were higher than those of the upper parts, whereas the thickening of the middle and lower branches of R766 was more pronounced than that of R2928, which may be part of the reason for the open-type distribution of the branches.

Further comparison of the filling characteristics of rice grains at different panicle positions (Table 1 and Table 2) showed that spikelets per panicle, spikelets per panicle and 1000-grain weight at each panicle position of R2928 were significantly higher than those of R766, which was related to the genotypic characteristics of the variety; however, variation in the 1000-grain weight of R766 grains at different panicle positions was relatively small. The 1000-grain weight of the middle and lower panicles decreased by 1.55% and 0.38%, respectively, compared with that of the upper panicles, whereas that of R2928 decreased by 5.99% and 7.12%, respectively. Comparison of the grain setting characteristics of the two different panicle types of rice at different positions revealed that the grain filling of the middle and lower panicles in conventional panicle-type rice significantly decreased, whereas the blighted grain rate and the percentage of empty spikelets increased. The grain filling of the open-type rice R766 in the middle and lower panicles was the same as that in the upper panicles, and the setting rate and grain filling of the middle and lower panicles were significantly higher than those in R2928. In terms of the seed setting rate and grain-filling degree at the lower panicle position, the percentage of empty spikelets at the lower panicle position of R766 was more than 10 percentage points lower than that of R2828, and the grain filling was 24.75% higher than that of R2928. The superior filling and fruiting characteristics of the lower part of the panicle of R766 may be related to thicker primary branches in the middle and lower parts. In this experiment, under the same planting conditions, other than panicle type there were no significant differences in plant type-related factors, such as plant height, panicle length and stem length, between the open-type rice R766 and the conventional panicle-type rice R2928 (Table 3). Two years of consistent experimental patterns were also noted.

### 2.2. Analysis of the Causes of Branch Angle of Open Panicle-Type Rice

The greater angle of the branch and smaller curvature of the panicle in R766 may be attributed to its higher mechanical strength. Therefore, the stem and branch bending resistances, anatomical structures, and chemical compositions of R766 and R2928 were compared and analyzed separately.

#### 2.2.1. Mechanical Strength Analysis

R766 exhibited high mechanical strength in terms of the stem and rachis (Figure 3), with a stem bending resistance of 9.79 N, which was 50.38% higher than R2928. The bending resistances of the rachis and the middle and lower branches were significantly higher than that in R2928, with increases of 63.24% (rachis), 47.37% (middle primary branch), and 65.22% (lower primary branch), respectively. No significant differences were observed in the bending resistance of the upper primary branch of the panicle. The mechanical strength of the primary branch in the middle and lower parts of open panicle-type rice was significantly higher than that of conventional panicle-type rice, whereas the mechanical strength of the primary branch in the upper part of the panicle was equivalent. This high mechanical strength may explain the large angle of the primary branch in the middle and lower parts of the open-type rice panicles.

#### 2.2.2. Anatomical Structure Analysis

To investigate the mechanism of high mechanical strength in the middle and lower primary branches and stems of open-type rice, microscopic observations were conducted on the stems, rachises, and middle and lower primary branches of R766 and R2928. In addition to the significant differences in the diameter of the panicles, there were also significant differences in the microstructure of the panicle slices (Figure 4 and Figure 5). Except for the absence of a middle air cavity in the middle stem of the panicle, both rice types had distinct middle cavities in other parts. R766 had a thicker transverse stem wall and a smaller air cavity in the stem and vascular bundles. The number, cross-sectional area, and distribution of the vascular bundles differed significantly between the two rice panicle types.

Stem and primary stem wall thickness as well as the number of large and small vascular bundles differed significantly among the different panicle types of rice. The wall thickness of R766 and number of large and small vascular bundles were greater than those of R2928 (Figure 6). The wall thickness and number of large and small vascular bundles in the stem were significantly higher than those of R2928, with increases of 35.95% (wall thickness), 29.30% (number of large vascular bundles), and 35.88% (number of small vascular bundles). The wall thickness and number of small vascular bundles in the panicle were significantly or extremely significantly higher than those in R2928. In the rachis, this was 10.59% and 28.80% higher, respectively, and in the primary branch, it was 16.56% and 24.29% higher, respectively. No significant difference in the number of large vascular bundles in the panicle was observed compared to R2928.

Anatomical sections revealed that the number of large and small vascular bundles in R766 was significantly higher than that in R2928, and its individual vascular bundle area was also larger than that in R2928. Comparing the single stem large and small vascular bundle areas of R766 and R2928, we found that the large and small vascular bundle areas of open panicle-type rice were significantly higher than those of conventional panicle-type rice, and that the ratio of large to small vascular bundle areas was significantly lower than that of conventional panicle-type rice (Figure 7). Compared to R2928, the large vascular bundle area in the stem of R766 was 78.26% higher, and the small vascular bundle area was 150.00% higher. The ratio of large to small vascular bundle areas was 17.12% lower than that of R2928. The vascular bundle characteristics of the rachis exhibited the same pattern among different rice panicle types. Compared to R2928, the large vascular bundle area in the rachis of R766 was 84.21% higher, and the small vascular bundle area was 80.88% higher. The area ratio of the large to small vascular bundles in the rachis was 24.63% lower than that of R2928. Compared with R2928, the large vascular bundle area in the primary branch was 1.5-fold higher, the small vascular bundle area was 2.0-fold higher, and the ratio of size to vascular bundle area was 24.31% lower. Thus, compared with conventional panicle-type rice, although open panicle-type rice has a larger number and area of large and small vascular bundles, the relative number of small vascular bundles is higher. Moreover, the area of vascular bundles and the number of small vascular bundles in the primary stem of open panicle-type rice showed a greater increase.

#### 2.2.3. Chemical Composition Analysis

Owing to the greater wall thickness and diameter of the stem and panicle of R766, the wall areas of its stem and primary branch were significantly larger than those of R2928. Stem and panicle cellulose content differed significantly between open panicle-type and conventional panicle-type rice, whereas there was no significant difference in stem lignin content (Figure 8). Compared with R2928, the cellulose content in the stem of R766 was 7.51% higher, and there was no significant difference in the cellulose content between the rachis and primary branch. No significant difference was observed in the lignin content of the stems and primary branches of the different panicle types of rice, and the lignin content in the rachis was 8.03% higher than that of R2928.

In summary, open panicle-type rice demonstrates a significant improvement in the mechanical strength of the stem, rachis and primary stem compared to the conventional panicle type, mainly reflected in the increase in wall thickness of its anatomical structure and the increase in vascular bundles, especially the increase in the proportion of small vascular bundles, which in turn increases its mechanical strength. The differences between the cellulose and lignin contents and their mechanical strengths were not significant.

### 2.3. Analysis of the Causes of Panicle Phenotype

#### 2.3.1. Correlation between Mechanical Strength Structure, and Composition

There was a significant correlation between the mechanical strength of the stems, rachises, and primary branches and their anatomical structures and components. Differences occurred in the correlation between the mechanical strength of different parts and anatomical structures and components (Figure 9). The bending resistance of the stems was positively correlated with their structure, with a significant positive correlation with the diameter, number, and area of large vascular bundles (Figure 9A). Stem diameter and the number and area of large and small vascular bundles were significantly positively correlated. There was a significant or extremely significant positive correlation between the numbers and areas of large and small vascular bundles. No significant correlation was observed between stem lignin content and mechanical strength; however, there was a significant positive correlation of cellulose with lignin content.

The bending resistance of the rachis was positively correlated with its anatomical structure and composition, with a significant or extremely significant positive correlation with the diameter of the rachis, the area of large and small vascular bundles, the number of large vascular bundles, and the content of lignin and cellulose (Figure 9B). At the same time, a highly significant positive correlation of the diameter of the rachis with the number of large vascular bundles, as well as with the area of large and small vascular bundles, was found. There was also a highly significant positive correlation between the number and area of large and small vascular bundles. Lignin and cellulose contents were significantly positively correlated and there was a significant or extremely significant positive correlation between lignin and cellulose content and the areas of large and small vascular bundles.

Bending resistance of the primary branch was positively correlated with its anatomical structure and negatively or not at all with its composition (Figure 9C). There was a significant or extremely significant positive correlation between the diameter, number of small vascular bundles, and areas of large and small vascular bundles. There was a highly significant positive correlation between the diameter of the primary branch and the number of small vascular bundles, as well as between the area of the large and small vascular bundles. There was a significant or extremely significant positive correlation between large and small vascular bundles in terms of their quantity and area. There was a significant or extremely significant negative correlation between the cellulose content and the anatomical structure of the primary branches, whereas there was no significant correlation between the lignin and cellulose content and the anatomical structure of the primary branches.

#### 2.3.2. Correlation between Branch Angle and Structure at Different Spike Positions

The angle of the upper panicle branch of the open panicle-type rice was not significantly different from that of the conventional panicle type, and its mechanical characteristics and components were also not significantly different from those of the conventional panicle type. Therefore, there was no significant correlation between the angle of the upper panicle position of open panicle-type rice and the panicle structure. The angle of the primary branch in the middle and lower parts of the panicle was significantly positively correlated with panicle structure (Figure 9D). The angle of the primary branch in the middle and lower parts of the panicle was significantly positively correlated with its resistance to bending, and the number and area of large and small vascular bundles. An increase in the diameter of the primary branch and the number and area of large and small vascular bundles significantly enhances its bending resistance, which may increase the angle of the primary branch.

#### 2.3.3. Correlation between Different Panicle Setting Rates and Panicle Traits

The setting rates of the middle and lower panicles in open panicle-type rice were significantly higher than those in conventional panicle-type rice. By analyzing the factors affecting the setting rate of the middle and lower panicles, we found that there was a significant positive correlation between the setting rate, full grain rate, and panicle structure of the middle and lower panicles, whereas there was a significant negative correlation between the withered grain rate and panicle structure (Figure 10). The setting rate of the middle panicle position was significantly positively correlated with the angle of the middle and lower panicle branches, diameter of the rachis, and area of the large vascular bundle of the primary branch. The setting rate of the lower panicle position was significantly or extremely significantly positively correlated with the branch angle, rachis diameter, large and small vascular bundle areas, primary branch diameter, and large and small vascular bundle areas in the middle and lower panicle positions. The full grain rate of the middle and lower panicle positions showed a similar correlation with the aforementioned panicle traits, and the positive correlation was stronger. In contrast, there was a significant or extremely significant negative correlation between the withered grain rate at different panicle positions and the aforementioned panicle traits. Thus, increasing the angle of the branches in the middle and lower panicles can significantly improve the seed setting rate and grain fullness in the middle and lower panicles, which is beneficial for increasing the weight of single panicle grains and thereby increasing rice yield.

## 3. Materials and Methods

### 3.1. Experimental Materials and Design

R766 was created through natural mutation, followed by hybridization and backcrossing, to create an open panicle-type germplasm resource (Figure 1). Preliminary comparison showed that open panicle-type rice exhibited a large angle of branches and an “open-type” panicle type while ensuring the number of grains per panicle. Therefore, this experiment used conventional panicle rice R2928 and open panicle rice R766 and was carried out from 2021 to 2022 at the agricultural garden experimental base of southwest university of science and technology (31°32′ N, 104°41′ E). The 2-year sowing dates were between 10 and 15 April, and the planting period was between 8 and 12 May. One seedling was planted per hole, and the row and plant spacing was 0.33 × 0.165 m. The soil was purple rice soil, with a basic nutrient content of 1.95 g/kg of total nitrogen, 79.8 mg/kg of available nitrogen, 42.8 mg/kg available phosphorus, and 75.8 mg/kg available potassium. The field cultivation management measures for the entire growth period were the same as the general field production standards.

### 3.2. Test Methods

During the flowering period, the simultaneous flowering of rice panicles was marked, and 10 labeled rice panicles were taken from each variety 30d after flowering. The distribution of grains at different positions, filling, and fruiting characteristics of different varieties, as well as the branch angle, bending resistance, anatomical structure, and cellulose, and lignin content, were examined.

#### 3.2.1. Yield and Yield Components

The rice ears were divided into three parts according to the number of branches, namely the upper part of the ear, the middle part of the ear and the lower part of the ear from top to bottom (when the number of branches cannot be divided evenly, the number of upper and lower ear positions should be equal). Each ear was counted and then sampled. At maturity, rice grains were collected from a 20 m^2^ sampling area in each subplot and used to calculate grain yield, which was then adjusted to 13.5% safe moisture content by weight. The grain yield components were measured, including the 1000-grain weight, which was calculated from three replicates; grain filling and panicles (containing the number of full grains and the number of empty grains) were measured.

#### 3.2.2. Branch Angle Analysis

The rice panicles were divided into three parts (upper, middle, and lower) based on the number of primary branches. The branch angles and bending resistance of these three parts were calculated separately after removing the grains. For the branch angle, a protractor was used to measure the angle between the primary branch and the rachis of the panicle in different parts. An HP-5 thrust gauge was used to measure the bending resistance of the branches. The reading of the thrust gauge when the middle of primary branch was deformed by 45° was recorded as the bending resistance of the branch.

#### 3.2.3. Measurement of Vascular Bundle Area

Marked plants were randomly selected from the field, and the primary branches, neck nodes, and stem nodes were collected at a height of 20 cm above the ground. Paraffin sections were prepared and placed in the Mshot digital microscopic analysis system for cross-sectional photography to observe the size and quantity of vascular bundles in the various parts.

#### 3.2.4. Determination of Cellulose and Lignin Content

Randomly selected marked plants in the field were blanched at 105 °C for 30 min and dried at 80 °C for 48 h until a constant weight was achieved. The stems, rachises, and primary branches were crushed separately using a Pulverisete14 variable-speed high-speed rotary crusher and then passed through an 80-mesh sieve. The cellulose and lignin contents were determined according to the operating manual of the FibeBtecTMM6 1020/102 cellulose analyzer from FOSS.

### 3.3. Statistical Analysis

Data were analyzed using analysis of variance (ANOVA), and means were compared based on the least significant difference (LSD) test at the 0.05 probability level by using SPSS 25.0 (Statistical Product and Service Solutions Inc., Chicago, IL, USA). Origin Pro 2020 (OriginLab, Northampton, MA, USA) was used to generate figures.

## 4. Discussion

### 4.1. Factors Affecting Panicle Traits in Rice

The number of tillers, grains, seed setting rate, and 1000-grain weight are the main factors affecting rice yield, and the key indicators of grain number, seed setting rate, and 1000-grain weight depend on rice panicle traits, such as panicle length, number of primary and secondary branches, and grain number per panicle. These traits, also known as the panicle structure, are key factors affecting rice yield [13]. Studies have shown that panicle traits are inherited in a quantitative manner with a complex genetic basis, controlled by a large number of quantitative trait loci (QTLs) and influenced by environmental factors. Hundreds of QTLs have been reported previously [14]. Although many QTLs related to panicle structure have been identified, fewer QTL genes have been cloned. Loose panicles are a typical genetic mutation in rice. Currently, a total of seven genes controlling the loose panicle trait in rice have been reported, namely *spr1*, *spr2*, *spr3*, *spr4*, *spr5*, *spr8*, and *OsLG1*. Among these, *spr1* and *spr8* are controlled by one recessive nuclear gene, whereas the other five genes for loose panicle traits were controlled by dominant genes. Among the genes reported to control loose panicle traits, only the dominant genes *spr3* and *OsLG1* have been cloned [15,16]. Regarding the panicle phenotype of loose panicle traits, some researchers used γ radiation to induce a loose panicle mutant *ET2* in *japonica* rice varieties [17]. The angle of primary branch attachment at the lower part of the panicle was approximately 30°–40°, and the angle of primary branch attachment was significantly increased. The entire panicle resembled that of an open panicle. Other researchers have also discovered a *japonica* rice mutant *sp* with a larger stem angle, named this panicle type “loose panicle type”, and located it on chromosome 4 [18]. This was also observed in wild-type rice. The *OsLG1* gene controls the characteristics of loose panicle branches, mainly by enlarging the base of the branches, leading to an increase in the angle of the branches [15]. Although there are fewer molecular biology studies on the “open panicle” or “loose panicle” angle of major branches in China compared to the upright panicle type, a relatively complete technical system has also been formed [19]; however, current research on loose-panicle rice mainly focuses on *japonica* rice, and the best panicle type proposed in this research is the upright panicle type. This upright panicle type, controlled by the *DEP2* gene, is increasingly being studied as an important indicator of the ideal plant type in the *japonica* rice region of northern China [20]. However, although upright panicles can improve population permeability, the increase in grain density of upright panicle rice increases the humidity of the panicle layer, especially in the southwestern *indica* rice region of China, where high humidity and low sunlight are the main ecological conditions, which can lead to a high incidence of diseases, such as rice blast and rice smut [21].

So far, there has been no research on open panicle-type mutants or genes derived from *indica* rice. The *indica* open panicle-type variety created in this study not only improved panicle permeability, but also reduced grain density per unit space, which may contribute to improving the population ecology of rice in the southwestern *indica* rice region of China.

### 4.2. Physical and Chemical Effects of Panicle Traits

The vascular bundle is a transport channel for photosynthetic substances, water, and minerals in rice. The size and quantity of vascular bundles in the panicle directly affect the rate of photosynthetic production like a “sink”. Extensive research has shown a significant positive correlation between the number and size of vascular bundles in rice panicles, and panicle traits [18,22]. Usually, the grains in the rice panicle are divided into strong and weak grains, and poor filling of weak grains in the lower part of the panicle is the main reason for the decrease in rice yield. The main reason for poor filling of weak grains is insufficient supply of assimilation products [23,24]. Further research on the relationship between the vascular bundle characteristics and the filling degree of strong and weak potential grains revealed that an underdeveloped vascular system of weak potential grains is an important reason for the increase in the filling degree of weak potential grains [25]. A comparison of the contributions of small and large vascular bundles to the transportation of nonstructural carbohydrates in stems showed that small vascular bundles may contribute more to the transportation of nonstructural carbohydrates after flowering, which is conducive to the accumulation of panicle substances [26]. There are significant differences in the development, diameter, and number of vascular bundle sieves in different parts of the rice panicles among different rice varieties [27,28]. A large number and area of vascular bundles are key indicators of ideal panicle design and breeding [29].

Plant cell walls are composed mainly of cellulose and lignin, which form cross-linked network structures. The basic units of cellulose are fiber filaments, which are the main source of mechanical support for the non-lignified cell walls in rice. The presence of cellulose can increase the mechanical strength of plants, while maximizing their toughness. A reduction in cellulose content can lead to brittle stems [30]. Lignin is located between cellulose microfibrils in plant cell walls and combines with hemicellulose to form a three-dimensional network structure that can play a compressive role and improve the mechanical properties of plants [31]. The higher the cellulose and lignin content in a certain volume of rice tissue, the greater the mechanical strength of the rice and the stronger its ability to support rice stems, panicles, and primary branches [32]. Lignin also increases cell wall thickness and improves the overall mechanical strength of plants, and its content is related to the overall material transport capacity of vascular bundles [33].

The two different panicle types of rice varieties in this experiment did not have brittle stems or mutants with high or low cellulose and lignin content. Therefore, the differences in cellulose and lignin content in the different parts were relatively small, resulting in little correlation between branch angle and cellulose and lignin content. The open panicle-type rice variety R766 had significantly higher numbers and areas of large and small vascular bundles in the stem, rachis, and primary stem than conventional panicle-type rice. In particular, the areas of large and small vascular bundles in the panicle branches were more than twice that of conventional panicle-type rice. The larger the area of the large and small vascular bundles, the thicker the stem wall and the greater the mechanical and supporting forces of each part of the panicle, making the entire ear of R766 upright and the primary branch stronger. In contrast, the smaller the area of the vascular bundles, the smaller the mechanical and supporting forces on each part of the panicle, resulting in overall bending and drooping of the panicle in conventional rice.

### 4.3. The Formation Mechanism of Panicle Structure in Open Panicle Rice and Its Agronomic Significance

There were significant differences in the mechanical structure and strength of the stem, panicle, and primary branch between the open panicle-type rice R766 and conventional panicle-type rice R2928. The stem wall thickness, large and small vascular bundles on the transverse section of the stem, and large area of the open panicle-type rice R766 resulted in greater mechanical strength in the rachis and stem than in conventional rice (Figure 11). This may explain the open type of the panicles; however, the multiple vascular bundles and high mechanical strength of the stem are key factors in improving the plant’s lodging resistance, which has been widely studied and confirmed [34,35,36]. We believe that the high mechanical strength of the rachis and branches is the main reason why their panicle remained firm during the filling period (Figure 10). The mechanical strengths of the rachis and stems of the conventional rice were relatively low. As the weight of the spikelet grains increases, the panicles cannot maintain a straight shape during the filling period. The high mechanical strength of the rachis in open panicle-type rice maintains the relatively light curvature of the panicle, whereas the high mechanical strength of the branch stems maintains the middle and lower spikelets in the upright and extended states. The author observed swelling at the angle between the primary branch and the rachis of different panicle types of rice, and an increase in cell width at the swelling area led to an increase in the angle of branch growth.

Research has shown that earlier-flowering superior spikelets, usually located on apical primary branches, fill fast and produce larger and heavier grains. On the other hand, later-flowering inferior spikelets, usually located on proximal secondary branches, are either sterile or fill slowly and produce smaller grains [37,38]. Compared to superior spikelets, inferior spikelets bloom later and start filling more slowly, ultimately resulting in a lower 1000-grain weight [36]. The 1000-grain weight and seed setting rate of these superior and inferior spikelets are more pronounced in some heavy-panicle super rice varieties, and improving the filling characteristics of grains in the lower part of the panicle is an effective way to further increase rice yield. Currently, plant-type breeding has become an important direction in super rice breeding [39,40,41,42], and panicle type, which is an important indicator of plant type, is widely used in breeding selection [14,43,44]. At present, there are many studies on the erect panicle type of rice [45,46,47] as an *indica* rice variety and loose-panicle rice R766 has a large number of vascular bundles in various parts and a large area (especially in the middle and lower branches of the panicle), which can ensure efficient and sufficient transportation of assimilates to the grains in the middle and lower parts of the panicle. This is conducive to the filling of grains in the middle and lower parts of the panicle, laying the foundation for increasing the rice yield and improving the quality of grains in the lower part of the panicle. Simultaneously, during the filling period, the primary branches were scattered in the field and it can reduce the shading of the panicle layer, which not only reduces the humidity of the panicle, but also the incidence of diseases such as rice smut. Simultaneously, the light intercepted by the panicles is lower and, thus the light that reaches the leaves increases during the flowering and filling periods, which can guarantee light energy for the efficient development of post-flowering photosynthesis in rice populations. It can also optimize the population microclimate of high-density rice production systems, such as machine insertion and live streaming.

## 5. Conclusions

This study compared the differences in panicle structure, mechanical properties, yield, anatomical characteristics, and chemical composition between the open panicle-type *indica* rice R766 and curved-panicle type rice R2928. It was found that the angle of the middle and lower branches of open panicle type *indica* rice R766 was above 35°, significantly higher than that of conventional panicle-type rice varieties. The stem wall thickness, cross-sectional size, and large number and area of vascular bundles of R766 make it more mechanically strong than the panicle axis and stem of conventional rice. The filling and fruiting characteristics of the grains in the middle and lower panicles of R766 were basically the same as those in the upper part and were significantly better than conventional panicle-type rice varieties.

## Figures and Tables

**Figure 1 plants-13-02035-f001:**
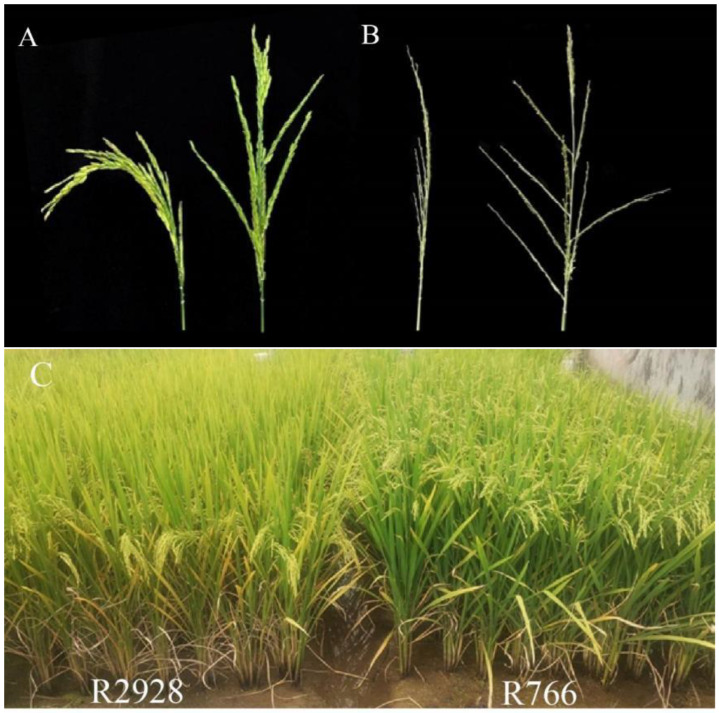
Phenotype of open panicle-type rice. (**A**,**B**): phenotypes of panicle branches ((**A**): with grains, (**B**): without grains). (**C**): Field phenotype during grain-filling period (R2928: curved-panicle type, R766: open panicle type).

**Figure 2 plants-13-02035-f002:**
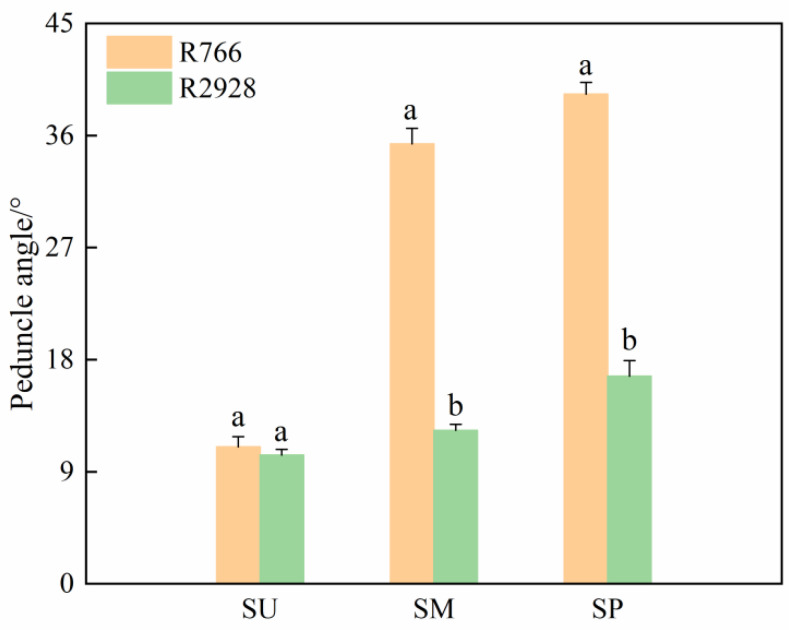
Analysis of panicle phenotypes for open panicle-type rice. SU: Upper part of panicle, SM: middle part of panicle; SP: lower part of panicle. The same column of different lowercase letters indicates that there is a significant difference at the 5% level between the same panicle position and different varieties.

**Figure 3 plants-13-02035-f003:**
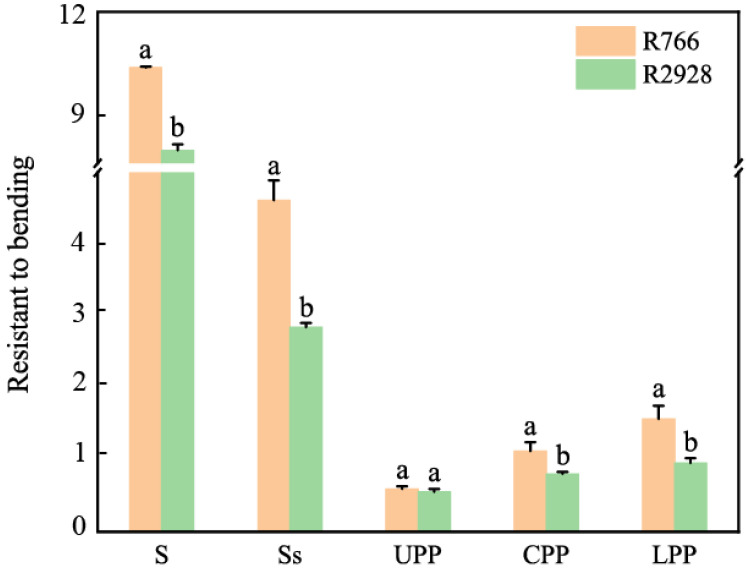
Bending resistance of different parts. S: Stem, Ss: rachis, UPP: upper primary branch, CPP: middle primary branch; LPP: lower primary branch. The same column of different lowercase letters indicates that there is a significant difference at the 5% level between the same panicle position and different varieties.

**Figure 4 plants-13-02035-f004:**
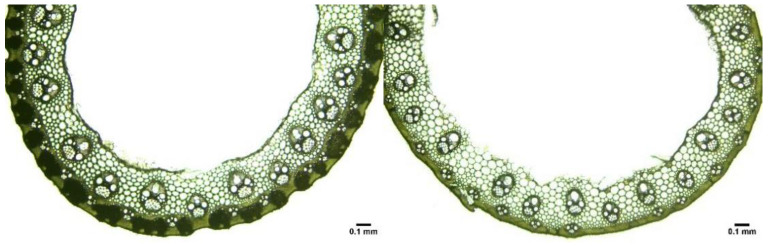
Microscopic images of R766 and R2928 slices.

**Figure 5 plants-13-02035-f005:**
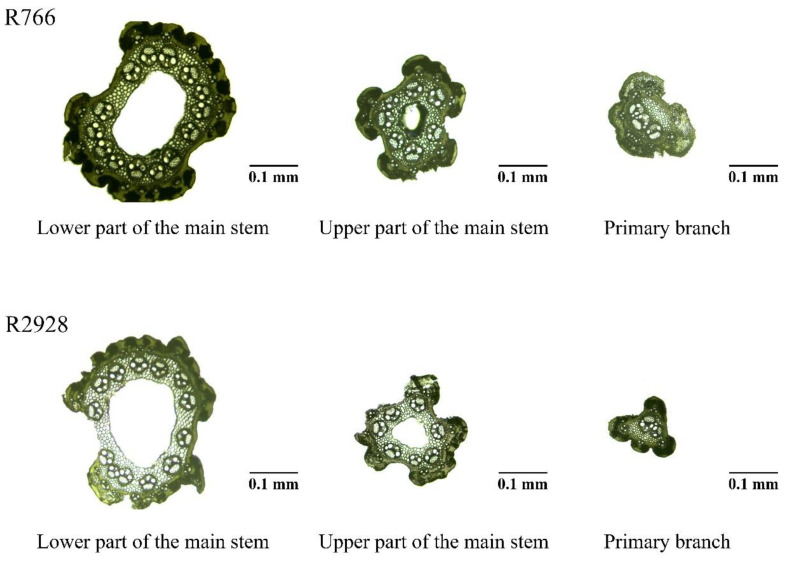
Cross section of stems and primary branches of different panicle types of rice.

**Figure 6 plants-13-02035-f006:**
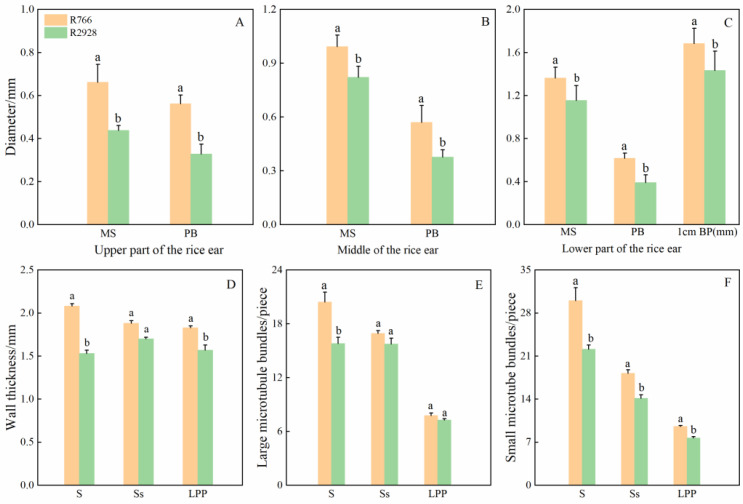
Comparison of cross-sectional structure of rice panicles with different panicle types. (**A**–**C**) Comparison of cross-sectional diameters of different panicle types of rice; (**D**–**F**) Comparison of stem wall thickness and size of panicle cross-section of rice with different panicle types, and the number of vascular bundles) MS: main stem, PB: primary branch, 1 cm BP (mm): 1 mm below the panicle stem node. The large vascular bundle refers to the vascular bundle distributed in the thin-walled tissue inside the stem epidermis, while the small vascular bundle refers to the vascular bundle distributed on the surface of the stem epidermis. Same below. S: stem, Ss: rachis; LPP: lower primary branch. The same column of different lowercase letters indicates that there is a significant difference at the 5% level between the same panicle position and different varieties.

**Figure 7 plants-13-02035-f007:**
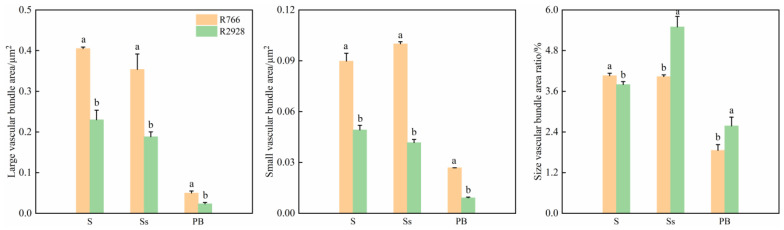
Comparison of vascular bundle area of stem and primary branch of different panicle types of rice. Note: S: stem, Ss: rachis; PB: primary branch. The same column of different lowercase letters indicates that there is a significant difference at the 5% level between the same panicle position and different varieties.

**Figure 8 plants-13-02035-f008:**
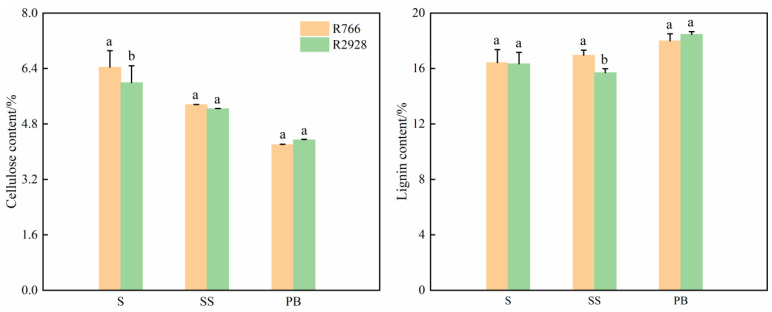
Comparison of main chemical components of stems and primary branches of different panicle types of rice. S: stem, Ss: rachis; PB: primary branch. The same column of different lowercase letters indicates that there is a significant difference at the 5% level between the same panicle position and different varieties.

**Figure 9 plants-13-02035-f009:**
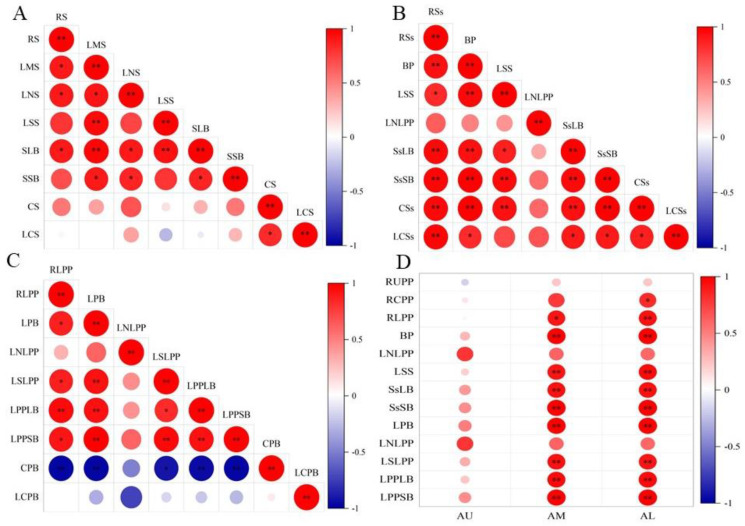
Correlation between bending resistance, branch angle, composition, and spike structure in different positions. ((**A**–**C**) The bending resistance of different parts is correlated with the structure and composition of microtubule bundles; (**D**) Correlation analysis between branch angle and panicle structure at different panicle positions). RS: stem bending resistance, RSs: rachis bending resistance; RLPP: bending resistance of the lower primary branch. LMS: stem diameter, BP: panicle node diameter, LPB: lower primary branch diameter, LNS: large vascular bundles of stem, LSS: small vascular bundles of stem, LNLPP: large vascular bundles of the lower primary branch, LSLPP: small vascular bundles of the lower primary branch, CS: stem cellulose, CSs: rachis cellulose, CPB: primary branch cellulose, LCS: stem lignin, LCSs: rachis lignin, LCPB: primary branch lignin, RUPP: upper primary branch stem bending resistance, RCPP: the bending resistance of the central primary branch, RLPP: bending resistance of the lower primary branch, SsLB: area of large vascular bundles in rachis, SsSB: area of small vascular bundles in rachis, AU: angle of the primary branch for upper panicle, LSLPP: number of small vascular bundles in primary branch, LPPLB: large vascular bundle area of primary branch, LPPSB: small vascular bundle area of primary branch. AM: angle of the primary branch for middle panicle; AL: angle of the primary branch for lower panicle. * and ** represent significant levels of 0.05 and 0.01, respectively.

**Figure 10 plants-13-02035-f010:**
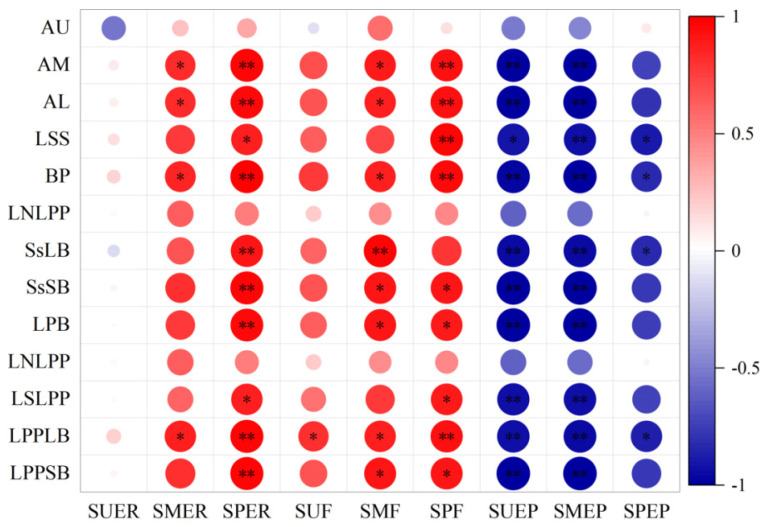
Correlation analysis between different panicle positions, seed setting rates and panicle structure. SUER: upper panicle setting rate, SMER: middle panicle setting rate, SPER: lower panicle setting rate, SUF: upper panicle full grain rate, SMF: middle panicle full grain rate, SPF: lower panicle full grain rate, SUEP: upper panicle withered grain rate, SMEP: middle panicle withered grain rate, SPEP: lower panicle withered grain rate, AU: angle of the primary branch for upper panicle, AM: angle of the primary branch for middle panicle, AL: angle of the primary branch for lower panicle, BP: rachis diameter, LSS: number of large vascular bundles in rachis, LNLPP: number of small vascular bundles in rachis, SsLB: area of large vascular bundles in rachis, SsSB: area of small vascular bundles in rachis, LPB: primary branch diameter, LNLPP: number of large vascular bundles in primary branch, LSLPP: number of small vascular bundles in primary branch, LPPLB: large vascular bundle area of primary branch; LPPSB: small vascular bundle area of primary branch. * and ** represent significant levels of 0.05 and 0.01, respectively.

**Figure 11 plants-13-02035-f011:**
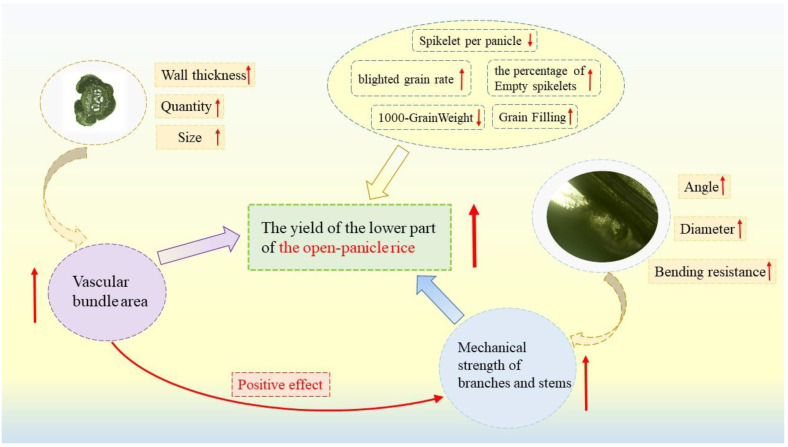
The formation mechanism of panicle structure in open panicle rice. ↑: Increase; ↓: reduce.

**Table 1 plants-13-02035-t001:** Grain-filling characteristics of different panicle positions in open panicle-type rice (2021).

Panicle Position	Variety	Spikelets per Panicle	1000-Grain Weight (g)	Grain Filling (%)	Blighted Grain Rate (%)	The Percentage of Empty Spikelets (%)	Grain Yield (kg/ha)
Upper	R766	74.57 ± 0.61 ab	23.86 ± 0.19 c	86.91 ± 0.99 a	2.69 ± 0.06 d	10.39 ± 0.03 b	2382.45 ± 27.02 ab
	R2928	79.71 ± 0.89 a	30.21 ± 0.25 a	83.56 ± 0.87 a	6.89 ± 0.03 bc	9.55 ± 0.04 b	2854.50 ± 33.30 a
Middle	R766	63.00 ± 0.96 b	23.49 ± 0.18 c	85.20 ± 0.92 a	4.70 ± 0.02 cd	10.11 ± 0.14 b	2355.15 ± 30.15 ab
	R2928	78.62 ± 1.12 a	28.40 ± 0.12 b	78.04 ± 0.85 b	13.14 ± 0.09 a	8.82 ± 0.06 b	1842.90 ± 24.65 c
Lower	R766	58.07 ± 0.14 c	23.77 ± 0.19 c	83.78 ± 1.15 a	7.09 ± 0.03 bc	9.13 ± 0.05 b	2109.75 ± 25.85 bc
	R2928	65.07 ± 0.21 b	28.06 ± 0.10 b	67.16 ± 0.71 b	10.1 ± 0.04 ab	22.74 ± 0.13 a	1885.65 ± 30.17 c

Lowercase letters show the significance of the differences among all the values in the same column (that is, both between genotypes and among panicle positions), in each column; values with the same letter are not significantly different (significance at *p* ≤ 0.05).

**Table 2 plants-13-02035-t002:** Grain-filling characteristics of different panicle positions in open panicle-type rice (2022).

Panicle Position	Variety	Spikelets per Panicle	1000-Grain Weight (g)	Grain Filling (%)	Blighted Grain Rate (%)	The Percentage of Empty Spikelets (%)	Grain Yield (kg/ha)
Upper	R766	73.29 ± 0.56 b	23.56 ± 0.32 b	85.44 ± 0.87 a	4.04 ± 0.05 e	9.52 ± 0.02 b	2319.45 ± 28.12 ab
	R2928	78.87 ± 0.42 a	28.96 ± 0.12 a	83.42 ± 0.63 a	7.30 ± 0.03 c	9.34 ± 0.03 b	2841.90 ± 24.43 a
Middle	R766	62.88 ± 1.34 cd	23.29 ± 0.13 b	84.58 ± 0.28 a	6.01 ± 0.03 d	7.07 ± 0.05 c	2303.25 ± 30.87 ab
	R2928	78.04 ± 0.89 a	28.25 ± 0.22 a	77.14 ± 0.35 b	14.28 ± 0.07 a	9.07 ± 0.08 b	1869.30 ± 21.76 c
Lower	R766	58.23 ± 0.18 d	23.17 ± 0.15 b	83.51 ± 0.67 a	8.88 ± 0.06 bc	7.61 ± 0.02 c	2098.05 ± 28.67 bc
	R2928	65.35 ± 0.25 c	28.05 ± 0.10 a	68.33 ± 0.42 c	10.57 ± 0.03 b	19.72 ± 0.09 a	1968.45 ± 21.44 bc

Lowercase letters show the significance of the differences among all the values in the same column (that is, both between genotypes and among panicle positions), in each column; values with the same letter are not significantly different (significance at *p* ≤ 0.05).

**Table 3 plants-13-02035-t003:** Differences in other agronomic traits of rice with different panicle types.

	2021 Year			2022 Year		
Variety	Plant Height (cm)	Panicle Length (cm)	Culm Length (cm)	Plant Height (cm)	Panicle Length (cm)	Culm Length (cm)
R766	115.40 ± 0.22 a	20.60 ± 0.24 a	86.92 ± 0.27 a	115.43 ± 0.41 a	20.9 ± 0.16 a	86.83 ± 0.08 a
R2928	115.90 ± 0.21 a	20.63 ± 0.20 a	87.47 ± 0.38 a	116.03 ± 0.54 a	20.51 ± 0.26 a	87.03 ± 0.42 a

In each column, values with the same letter are not significantly different (significance at *p* ≤ 0.05).

## Data Availability

The data presented in this study are available upon request from the authors.

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
