# Peer review of "Application of Open Panicle Traits in Improving the Filling Characteristics at the Base of Indica Rice Panicles"

_plants, 2024, doi:10.3390/plants13152035_

Round 1

Reviewer 1 Report

Comments and Suggestions for Authors

Dear Authors

This manuscript is difficult for reader to understand. Comments and examples are sown as follows.

1.      The authors should describe the "Materials and Methods" in as much detail as possible.

Example: How to divide the panicle into upper, middle and lower parts. How to calculate the grain yield (kg/ha).

2.      The Authors should show not only grain yield but also panicle number, culm length, panicle length, plant dry matter weight, panicle weight and plant height for R766 and 2928.

3.      The Authors should describe in text based on the data of “Table” and “Figure”.

Example 1: The angles of the middle and lower branches of R766 were 65.10% and 57.62% higher than those of R2928, respectively (Line 102-104). However, Figure 2 shows that the angles of the middle and lower branches of R766 were more than twice as higher than those of R2928, respectively.

Example 2; Further comparison of the filling characteristics of rice grains at different panicle positions (Table 1) showed that the number of grains and 1000-grain weight in the lower part of the rice panicle were lower than those in the upper part (Line 116-118). However, “Table 1” does not show that 1000-grain weight in the lower part of the rice panicle were significantly lower than those in the upper part.

4.      The Authors should show the values based on the number of significant digit in “Table”.

Example: The values of “number of spikelets per panicle” are shown two decimal places in Table 1. Authors counted the number of spikelets with natural number. Therefore, these values are in accordance with the number of significant digit. Authors should correct the values in accordance with the number of significant digit not only in text but also in “Tables”.

5.      The Authors should recalculate and correct the values in “Tables” and “Figures”.

Example: Significant difference at the 5% level between the same panicle position and different strains is shown the different letter in Table 1. However, there are inconsistencies in the notation of significant differences (Empty panicle rate; 8.82a and 9.13b). “Grain Filling rate (Grain Filling (%))”, “Shrunken grain rate” and “Empty grain rate (Empty panicle rate)” of lower position in R2928 are 68.33, 10.57 and 11.67, respectively. Therefore, the total of these percentages is 90.57% (not 100%).

6.      The authors should unify the terminology described in text, Figure and Table.

Example:

“Empty panicle rate (%)” in Table, “Empty grain rate” in text,

“Grain Filling (%)” in Table, “grain filling” or “the full grain rate” in text,

“Spikelet per panicle” in Table, “the setting rate” in text,

and et al.

Therefore, authors should rewrite the manuscript as a whole, as readers will be able to understand it clearly.

Author Response

Point 1: The authors should describe the "Materials and Methods" in as much detail as possible.Example: How to divide the panicle into upper, middle and lower parts. How to calculate the grain yield (kg/ha).

Response 1: Thank you for your comment.  We have already explained clearly it (line391-399).

Point 2: The Authors should show not only grain yield but also panicle number, culm length, panicle length, plant dry matter weight, panicle weight and plant height for R766 and 2928.

Response 2:  Thank you for your comment.  We have revised it(line144-152)

Point 3:The Authors should describe in text based on the data of “Table” and “Figure”.

Example 1: The angles of the middle and lower branches of R766 were 65.10% and 57.62% higher than those of R2928, respectively (Line 102-104). However, Figure 2 shows that the angles of the middle and lower branches of R766 were more than twice as higher than those of R2928, respectively.

Example 2; Further comparison of the filling characteristics of rice grains at different panicle positions (Table 1) showed that the number of grains and 1000-grain weight in the lower part of the rice panicle were lower than those in the upper part (Line 116-118). However, “Table 1” does not show that 1000-grain weight in the lower part of the rice panicle were significantly lower than those in the upper part.

Response 3: Thank you for your comment. We have revised it(line107,118-120,127,129,133,135)

Point 4:The Authors should show the values based on the number of significant digit in “Table”.Example: The values of “number of spikelets per panicle” are shown two decimal places in Table 1. Authors counted the number of spikelets with natural number. Therefore, these values are in accordance with the number of significant digit. Authors should correct the values in accordance with the number of significant digit not only in text but also in “Tables”.

Response 4: Thank you for your comment. We have revised it(line151,table2)

Point 5:The Authors should recalculate and correct the values in “Tables” and “Figures”.

Example: Significant difference at the 5% level between the same panicle position and different strains is shown the different letter in Table 1. However, there are inconsistencies in the notation of significant differences (Empty panicle rate; 8.82a and 9.13b). “Grain Filling rate (Grain Filling (%))”, “Shrunken grain rate” and “Empty grain rate (Empty panicle rate)” of lower position in R2928 are 68.33, 10.57 and 11.67, respectively. Therefore, the total of these percentages is 90.57% (not 100%).

Response 5: Thank you for your comment. We have corrected it in the revised manuscript(line 170).

Point 6:  The authors should unify the terminology described in text, Figure and Table.

Example:

“Empty panicle rate (%)” in Table, “Empty grain rate” in text,“Grain Filling (%)” in Table, “grain filling” or “the full grain rate” in text,“Spikelet per panicle” in Table, “the setting rate” in text,and et al.

Response 6: Thank you for your comment. We have revised it   

Reviewer 2 Report

Comments and Suggestions for Authors

The authors reported a highly detailed comparison between two rice lines with different panicle morphologies. They found that the line with an umbrella panicle (R766) yielded better than the standard curved panicle (R2928) in the middle and lower positions. Overall, the manuscript is interesting and R766 could be a valuable resource for breeding better rice for adaptation to the local environment. Here are my comments:

Line 75: Please include a short paragraph here summarizing the experimental work/plan and hypothesis. It is just to provide a better clarity to readers so they can understand what you are doing here before going into the results section.

Line 76-88: Please also introduce R2928 here and explain why R2928 is a good line to compare against R766. Does it have any commercial/research importance/relevance?

Line 409-442: Has there been any work done to identify the causative gene for the umbrella phenotype in R766 panicle? Or any future plan? Please elaborate on that.

I am not sure if it is formatting issue, but please check for spelling/formatting errors, such as "progragm" in Line 42, first half of the sentence in Line 58, "was" in Line 100, etc.

Author Response

Point 1: Line 75: Please include a short paragraph here summarizing the experimental work/plan and hypothesis. It is just to provide a better clarity to readers so they can understand what you are doing here before going into the results section.

Response 1: Thank you for your comment. This experiment used conventional panicle rice R2928 and umbrella panicle rice R766 as experimental materials. This study provides a reference for promoting the umbrella panicle-type indica rice variety to improve yield and disease resistance in ecological environments with high humidity and low sunlight.

Point 2: Line 76-88: Please also introduce R2928 here and explain why R2928 is a good line to compare against R766. Does it have any commercial/research importance/relevance?

Response 2: Thank you for your comment. R2928 is a high-yield rice with large promotion area and it has strong stress resistance in Sichuan.

Point 3:  Has there been any work done to identify the ? Or any future plan? Please elaborate on that.

Response 3: Thank you for your comment. We have considerated the causative gene for the umbrella phenotype in R766 panicle in the future.

Round 2

Reviewer 1 Report

Comments and Suggestions for Authors

Dear Authors

The authors revised the description (line 95-134) concerning the data shown in Figure 2 and Table 1. In addition, the authors added the data in Table 1 and 2. However, they did not revise the manuscript as a whole, according to reviewer’s comments. Furthermore, this manuscript still contained arguments that were not based on the results. The description of "Materials and Methods" was not added and revised for the reader to understand (Especially, How to divide the panicles into upper, middle and lower parts.). In addition, the descriptions in "Abstract" and "Results" were not corrected to be based on the data in "Table" or "Figure". Descriptions in “Table” and in the text were not corrected to values based on statistically significant. “Table” did not correct inconsistent data or incorrect statistical notation. Many incomprehensible descriptions have not been improved even in the revised manuscript. Therefore, this manuscript could be recommended to resubmit in “Plants”, after rewriting the manuscript as a whole.

Author Response

Point 1:  they did not revise the manuscript as a whole, according to reviewer’s comments.

Response 1: Thank you for your comment.  Based on your comments, we have revised the entire article.

Point 2:  The description of "Materials and Methods" was not added and revised for the reader to understand (Especially, How to divide the panicles into upper, middle and lower parts.).

Response 2: Thank you for your comment. We have clearly described the method of dividing parts:The rice ears were divided into three parts according to the number of branches, namely the upper part of the ear, the middle part of the ear and the lower part of the ear from top to bottom (when the number of branches cannot be divided evenly, the number of upper and lower ear positions should be equal); Each ear was counted and then sampled.(line 419-422).

Point 3:  In addition, the descriptions in "Abstract" and "Results" were not corrected to be based on the data in "Table" or "Figure".

Response 3: Thank you for your comment. The description of the Abstract and Results has also been corrected based on the data in the  "Table" or "Figure" (lines 17, 113, 146-148).

 Point 4: Descriptions in “Table” and in the text were not corrected to values based on statistically significant. “Table” did not correct inconsistent data or incorrect statistical notation.

Response 4: Thank you for your comment. We have re-analyzed the data in the table and have completed the revision (line 155,160)
